# EBR: Routing Protocol to Detect Blackhole Attacks in Mobile Ad Hoc Networks

**Deepika Kancharakuntla and Hosam El-Ocla ***

Department of Computer Science, Lakehead University, Thunder Bay, ON P7B 5E1, Canada
* Correspondence: hosam@lakeheadu.ca

**Abstract:** The presence of malevolent nodes in mobile ad hoc networks (MANETs) would lead to genuine security concerns. These nodes may disturb the routing process or deform the pattern of the data packets passing through the network. The MANET is extremely liable to attacks, owing to its characteristics of the network framework, such as the absence of infrastructure, moveable topology, and a centralized control unit. One of the most common attacks in MANETs is the blackhole attack. MANET nodes are susceptible to spectacular degradation of network performance in the presence of such attacks. In this regard, detecting or preventing deceitful nodes that will launch blackhole attacks is a challenge in MANETs. In this paper, we propose an Enhanced Blackhole Resistance (EBR) protocol to identify and resist nodes that are responsible for blackhole attacks. EBR can avoid congested traffic by passing the data packets through a safe route with the minimum RTT. The EBR protocol uses a combination of time to live (TTL) and round trip time (RTT), which is also called a TR mechanism, to detect the blackhole attacks. Our algorithm does not require any cryptographic or authentication mechanisms. Simulation results prove that EBR behaves better than other protocols in terms of throughput, end-to-end delay, packet delivery ratio, energy, and routing overhead.

**Keywords:** wireless; MANET; blackhole; attack; routing; round trip time; time to live; congestion



## 1. Introduction

A MANET is a self-configuring network of dynamic nodes connected through wireless links. Every node may act as being a router or host, and it can move freely in any direction [1]. Data communication is achieved by routing packets to the end nodes through intermediate nodes. Numerous routing protocols are designed for communication in MANETs, and they are classified into three major types: proactive, reactive, and hybrid protocols [2]. Ad hoc On-demand Multipath Distance Vector (AOMDV) is a reactive routing protocol [3]. AOMDV is an efficient protocol when compared with single-route mechanisms such as Ad-hoc On-demand Distance Vector (AODV), particularly in case of nodes mobility where the amount of dropped data augments.

On the other hand, MANETs are susceptible to security issues and attacks because of the ease of mobility and its nature of having no infrastructure. A blackhole attack is also known as a packet-dropping attack, and this greatly reduces the network performance. Mostly we have two types of blackhole attacks, which are single blackhole node or multiple blackhole nodes. When there are several blackhole nodes, they collaborate with one another to disturb communication, and this is called a cooperative blackhole attack [4–7]. In normal reactive routing protocols and when a source node is willing to send data to a destination node and does not have a path, it sends out route request packets (RREQ). The data packets will be forwarded to the destination upon receiving route-reply packets (RREP) from the intermediate nodes or the destination. Nonetheless, the blackhole node, when receiving an RREQ packet, sends back an RREP promptly with false information of having the shortest path to the receiving end with a quite large sequence number. This high sequence number indicates the freshness of the path. If the source node picks such a route having blackhole node(s), these malicious nodes will drop the data packet [8].

There are several solutions proposed to prevent the blackhole attacks. Security measures were added to existing protocols and cryptographic mechanisms were added to the packets [9,10]. MANETs have limited resources; however, most of these methods impose burdens such as delay and excessive energy consumption, which in turn would diminish the lifetime of the network. On the other hand, different trust schemes and confidence models have been proposed [11,12]. This is to enhance the security in MANETs where nodes can ensure a neighbor's confidence directly or through recommendations from other nodes in the network. There are several security methods, such as in [13–16]. Cryptographic mechanisms consume a lot of energy for computation; however, these methods may behave worse than those non-secured techniques when some attackers exist [17,18]. Providing a securing protection to the routing messages does not necessarily assure the ability to detect the suspicious nodes [19]. This motivated us to develop a mechanism that highly secures data packets, reduces the energy consumption, and enhances the quality of service (QoS) performance.

In this paper, we propose an Enhanced Blackhole Resistance (EBR) algorithm to detect and, therefore, prevent or at least mitigate the data packets from going through routes having malevolent nodes in mobile networks. EBR can be applied to any reactive protocol. To select the path that enhances the data delivery performance, EBR is better to be used with a multipath routing protocol, so we use it here with an AOMDV method to obtain optimized routes. Each node inspects its neighbors and gives them confidence values to identify the blackhole nodes. This algorithm provides a solution for both single and collaborative attacks, and it significantly shows excellent performance when compared with other protocols. Each node can determine the confidence level of neighboring nodes through employing a combination of time to live (TTL) and round trip time (RTT), and this in turn can determine the existence of the blackhole node(s). Also, EBR can detect the congestion status of a route using the RTT. We call this combination the TTL/RTT (TR) algorithm. In the EBR protocol, each node can send an RREQ test message to a dummy destination node. If the receiving node(s) reply with an RREP message within a limited RTT, this will be considered as a blackhole node, and the confidence level will be low and, therefore, the route will not be selected. Otherwise, if the RTT is long, the testing node will assume that the route has an additional blackhole attacker in one or more of its nodes, and, also, these intermediate nodes of such route might be congested. TTL should have a small value to only examine the neighboring nodes and also to dodge immersing the network in unwanted routing overhead traffic. In this regard, our contributions in this paper are:

- Propose TR algorithm utilizing TTL and RTT;
- Propose EBR protocol based on TR algorithm. EBR enhances the network performance. EBR is able to do the following main functions:

  - ➢ Detect blackhole attacker(s) in a transmission route,
  - ➢ Detect data congestion at intermediate nodes in a transmission route;

- Filter out routes returned by AOMDV to avoid blackhole attackers and congested nodes;
- Compare our EBR protocol with other comparative methods through simulation.

The structure of this paper is outlined as follows: Section 2 presents a literature review. Section 3 outlines the proposed protocol. Section 4 presents the simulation results. Section 5 elaborates the protocol complexity. Section 6 summarizes the conclusions.

## 2. Related Work

AODV is a commonly used reactive protocol. However, it is at risk to the blackhole node as being a critical security threat to routing protocols, as pointed out in [20]. Ochola et al. [21] concluded that AODV has a low performance when the selected route has blackhole nodes. Also, in [22,23], it was proved experimentally that the performance of the AODV protocol collapsed in the presence of a blackhole attack.

Dokurer et al. [24] amended the AODV mechanism to lessen the possibility of having a wicked or single-blackhole node. However, this mechanism is susceptible to cooperative

blackhole threats when two sly attackers are cooperating to cause the attack. Also, this protocol ignores the shortest path whenever there are no malicious nodes.

SAODV [25] is an improved AODV routing mechanism to protect AODV against attacks. In this protocol, AODV requires a supplement message, which is a fusion of digital signature with the use of the private key of the actual sender and a hash value of the hop count. This message adds an extra routing overhead cost, and this accordingly augments the energy consumption of nodes.

In [26], a blackhole detection method called the Cooperative Bait Detection Scheme (CBDS) was proposed. It mainly aimed to prevent blackhole nodes from beginning collaborative blackhole attacks in MANETs. Using a backward tracking method, malicious attackers are detected and prevented from taking part in the data routing. In this model, when there is an obvious degradation in the packet delivery ratio (PDR), an alarm is triggered to start the malicious node detection scheme again. PDR may degrade for any other reason, such as traffic problems; however, this algorithm will consider it as a result of the existence of malicious nodes.

The authors in [27] proposed a Security Using Pre-Existing Routing for Mobile Ad hoc Networks (SUPERMAN) protocol. This mechanism solves problems of node certification, medium access control, and secure transmission for MANETs. SUPERMAN combines routing with security. However, it has a lot of overhead in comparison with other protocols, particularly when the number of nodes increases.

To overcome the negative effect caused by the blackhole nodes, [28] developed an evolutionary self-cooperative trust (ESCT) mechanism that fulfills high throughput and PDR; however, ESCT increases the routing overhead cost and also extends the end-to-end delay.

In [29], the authors proposed a protocol that utilizes feedback such as ACK messages from the destination or the intermediate nodes back to the source node. This is to monitor the status of the neighboring nodes as being malicious or legitimate. This approach is complicated and even improves the performance insufficiently.

The authors of [30] used a new Receive Reply method that enables the source node to scrutinize the destination sequence number linked with the RREP packet. This method introduced a pre-RREP message to determine the good nodes and malicious ones. This is a general approach to detect any intrusion. This system likely collapses in the presence of multiple collaborative blackhole nodes.

In [31], the authors provided an enhanced version of the AODV protocol to mitigate the blackhole attack. Using VERIFY and CHECKVRF messages, the proposed approach stretches the standard AODV protocol for certification purpose. When the destination receives the CHECKVRF packet, it replies with the FINALREPLY message to guarantee the genuineness of the path. The main concerns of this method are obviously the extra routing overhead appended in addition to the excessive energy consumption.

In [32], the authors proposed a mechanism based on setting a security bit in the RREP message. This model assumes that the suspicious node has no anxiety about that validity. The source utilizes that path and sends its data packets if the security bit is still one. In any other case, it assumes that the path passes through a sly node and accordingly drops that RREP packet. In [33], the authors discussed the drawback of this method: this security bit assumption is unrealistic. This technique cannot exactly determine the blackhole nodes.

In [34], a DAPV mechanism is introduced to find single and multiple malicious nodes and also the nodes that do not behave normally. DAPV relies on two primary steps, including the log information of peers and the Merkle Hash Tree to check these logs without disclosing the privacy of the checked nodes.

The authors of [35] developed a Delay Tolerant Network (DTN) to deal effectively with connectivity and latency issues in wireless networks. DTN is prone to attacks such as blackhole and greyhole due to the limited connectivity it has. In these attacks, malevolent nodes purposely drop packets. Statistical-based Detection of Blackhole and Greyhole attackers (SDBG) was suggested to address these types of attackers. Nodes can evaluate

forwarding behaviors of other nodes based on the encounter record histories exchanged between them. This solution can work with various dropping probabilities and different number of attackers but still is not very efficient, as it consumes a lot of energy to exchange encounter histories and evaluate attackers.

On the other hand, path selection is a challenging aspect in heavy-load networks, where data traffic congestion occurs frequently. Various algorithms were proposed to use the channels' capacity to alleviate the number of data packets going through nodes to avoid having congestion. In [36], the authors suggested a method to guarantee a fair power distribution in wireless network channels for real-time applications. Some other methods considered network resources such as RTT and the bandwidth to control the network traffic transmission over links, such as in [37–39]. In [39], the authors have developed a routing protocol that selects the route based on residual bandwidth at the intermediate nodes and the destination. The main concern of this protocol is its extra overhead incurred.

Some other techniques proposed using neural networks [40] and fuzzy control [41]. These approaches are mainly time- and energy-consuming methods.

The authors of [42] have proposed a protocol to pass the data through reliable links with less traffic. This method is based on node connectivity, where it won't guarantee high performance when node speed increases.

There are several factors to evaluate a security routing protocol against the blackhole threat detection, particularly in MANET nodes that use battery-based power. It is necessary to make a compromise between these factors to propose an efficient protocol. These factors include throughput, end-to-end delay, energy consumption, and routing overhead. In this regard, we propose our algorithm, which outperforms the performance of other protocols.

## 3. Proposed Protocol

### 3.1. Problem Statement

MANETs are very vulnerable to blackhole attacks. Whenever the source wants to send data packets and does not have a path towards the destination, it broadcasts a route request packet (RREQ). Perhaps a blackhole attacker node on receiving the RREQ packet will sends a route reply (RREP) packet immediately with false information of having the shortest path. The sender likely selects this unsafe path and forwards data packets though blackhole node(s), which in turn would drop these packets. In addition to the route security issue, energy consumption, throughput, and routing overhead are also affected by going through such attacker node(s).

### 3.2. Proposed Solution

To detect and prevent blackhole nodes, we propose an Enhanced Blackhole Resistance (EBR) algorithm in mobile networks. Each node examines its neighbors and determines their confidence degree to identify the blackhole nodes. EBR protocol provides a solution for both single and collaborative attacks. EBR employs a combination of time to live (TTL) and round trip time (RTT) to determine the existence of the blackhole node(s) in every available path. This combination is called the TTL/RTT (TR) method. In the EBR algorithm, each node can send a RREQ test message to a dummy destination node. If the receiving node(s) reply with an RREP message, this will be considered as a blackhole node(s) and the confidence level will be low, and, therefore, the route will not be selected. TTL should have a small value to only examine the neighboring nodes and, also, to avoid overwhelming the network with overload traffic. RTT in case of a blackhole attacker is a short time because this attacking node wouldn't consume processing time, as the fake route is always available to receive the data packets and hence drop them. If the RTT is long, this indicates that the route nodes are possibly congested and should be avoided even if the path is safe.

### 3.3. Methodology

This mechanism introduces a new concept of Enhanced Blackhole Resistance (EBR), which detects the blackhole node and also avoids the congested route. We introduce a

slight addition to the original reactive protocols by storing the successful round trip times. Every network's node must test the performance of its surrounding nodes to detect if any act dishonestly, such as blackholes or those congested nodes.

Using our TR mechanism in Figure 1, the major components of the EBR protocol are listed below:

1.  Route Request Tests: Each node in the network periodically sends a Test_RREQ message to a dummy destination node. The time to live (TTL) of the Test_RREQ message has an arbitrary value *n*. Enlarging *n* would increase the number of nodes to be examined and, therefore, the source node will have a better idea of those malicious nodes in a certain path. However, this large *n* would create extra overhead traffic in the network that should be avoided. On the other hand, it would be more precious that each node just examine the suspicious nodes in its close neighborhood, so it is ideal to have the value of *n* <=3;

2.  Route Response: Only a maleficent node will respond to this Test_RREQ message. If the source node receives an RREP to its Test_RREQ from one of its neighbors, the source can confirm that the current route has blackhole node(s) and changes its trust level;

3.  Trust Levels: We have two types of trust levels: TRUST and THREAT, as shown in Algorithm 1. When a new node joins the network, its trust level is set to TRUST as a good node and, therefore, its confidence is set to +1. When any node responds to the Test_RREQ message, then its trust level is updated to THREAT. In this case, the procedures in Algorithm 2 should be followed;

4.  Confidence Levels: If the node is set as a THREAT, we have two further types of confidence levels: negative and zero. When a node thinks its neighbor is a blackhole node, then it gives that particular node $-1$ confidence. However, if the neighbor is thought to be a victim of a blackhole node, the confidence of that node becomes zero;

5.  Node Integrity Test: In case of a THREAT node, we test if a neighboring node is a malignant or a victim node. In doing this, we use our TR mechanism with different possibilities listed below. The TR mechanism includes both Algorithms 1 and 2. $RTT_i$ and $RTT_a$ stand for instant and average RTT, respectively. TTL can have the value *n*.

    A.  When $RTT_i \leq RTT_a$ and TTL value is small: In this case, we can locate exactly the position of the blackhole node(s) using the RREP message. The blackhole node is the neighboring node if *n* = 1. If *n* = 2, the neighboring node is given the confidence of zero, as it is most likely a victim of a blackhole node next to it. Increasing *n* would refer to more potential nodes that act falsely in the current route;

    B.  When $RTT_i \leq RTT_a$ and TTL value is large: In this case, the source node can recognize that there is a possibility of having blackhole node(s) in the path, and it is far away from the source node depending on the value of *n*. However, the location of the blackhole node(s) cannot be determined exactly. This path should be avoided in routing the data packets. The confidence level of the neighboring node is not changed. In addition, the neighboring node is not congested and, therefore, it can be used safely in other paths;

    C.  When $RTT_i > RTT_a$ and TTL value is small: This can locate the blackhole node(s) in a specific route, but most likely it is not the neighboring node(s). In other words, the attacker node(s) is farther away from the source node and responds late because one or more of the preceding nodes, in a specific path, is congested. Therefore, the source node can recognize that the neighboring node(s) are likely congested and the farther node(s) is/are blackhole node(s). However, the neighboring node can be considered in any other route, as it is congested but safe. This only can be accepted if other paths are suspicious as they go through one of the blackhole attackers;

    D.  When $RTT_i > RTT_a$ and TTL value is large: Location of the blackhole attacker is somewhere away from the source node. $RTT_i$ is large because of having

several congested nodes or due to the large value of *n*. Therefore, the whole path should be avoided.

6. Data Communications: As shown in Algorithm 3, EBR is a combination of a reactive routing protocol and our TR algorithm. Selected routes should consider the confidence values calculated using our TR method during the testing phase. When the source node wants to communicate, it will only consider the routes which have positive confidence levels and, also, are less congested. Zero confidence indicates that a node has a blackhole attacker next to it. Negative confidence indicates that this is a blackhole node.

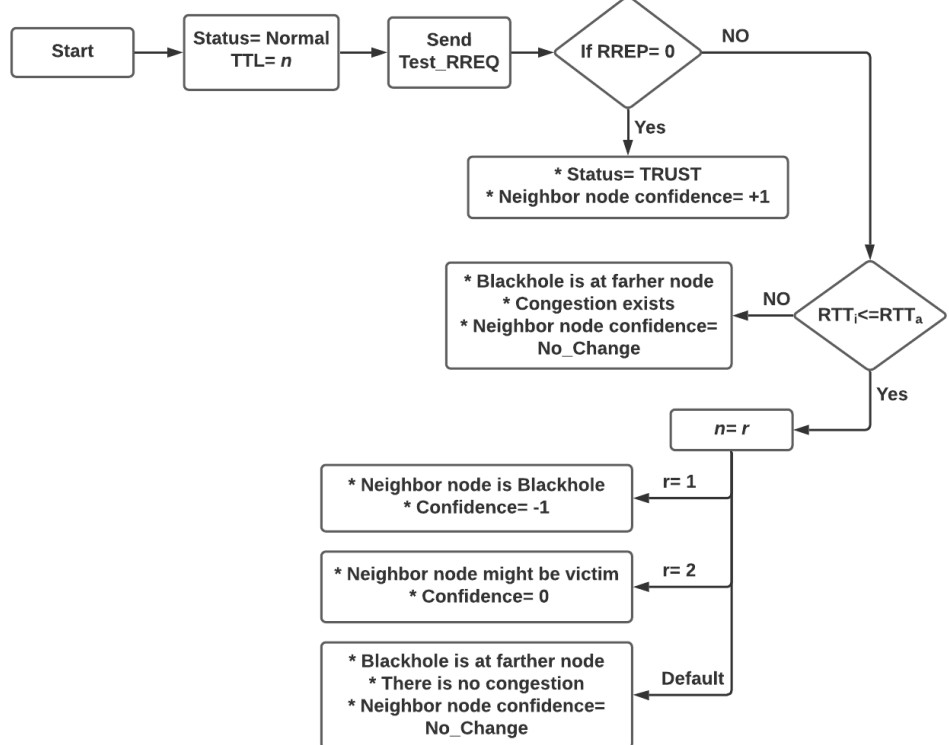

**Figure 1.** Flow chart of Enhanced Blackhole Resistance Method.

### 3.4. EBR Routes Properties

In algorithm 2, there are some cases where their confidence degrees have no change. Each node is supposed to run the integrity test periodically; therefore, confidence degrees will be decided for the close nodes to the source node. In other words, the blackhole node *W* is farther away from testing node *N*; however, *W* is close to another testing node *M*. Therefore, nodes in one route through *W* from *N* will have no change in their confidence degrees, but their confidence level will be different when *W* is tested through node *M*.

In Algorithm 3, array *A* will be sorted based on the minimum RTT of those routes obtained by the AOMDV protocol. Next, routes will be checked to find the one with a confidence higher than zero. If this is not available, array *A* is searched to find the route in which the confidence degree equals zero.

Some nodes might behave normally at some time and their confidence will be determined as TRUST; however, later these nodes will act maliciously. This can be detected if the testing node runs integrity tests more frequently; however, this would increase the traffic overhead. The mobility of nodes would negatively affect the RTT measurement and its accuracy, and this is one of the constraints of the RTT-based methods. There are plenty of protocols that use RTT in data communications of different systems, as outlined in [43]. RTT deterioration depends on types of the connection in the sense of being short-term or long-term connections. For long-term connections, RTT should be updated frequently to avoid performance degradation, especially in networks having high-speed-nodes mobility.

EBR uses the RTT to predict the congested routes, but this does not conflict with the congestion control protocols, such as NewReno, applied on nodes individually. RTT has been used over the years by researchers to evaluate network performance. RTT can possibly be measured via a few ICMP echo requests and response messages between the source and the destination nodes [44]. On the other hand, the mobility of nodes would result in links failure and packets loss. Multipath routing protocols such as AOMDV produce alternate routes available when communication disconnect occurs. EBR provides such alternate routes that are sorted based on the minimum RTT.

---

**Algorithm 1:** Test Neighbor Integrity

---

1. **Begin**
2. **Input** $n$
3. When a node joins the network
5.     Status = TRUST
6.     Confidence = +1
7. Broadcast Test_RREQ to neighboring nodes with TTL = $n$
8. **If** (RREP = 0)
9. **Then**
10.    No blackhole node exists
11.    Status = No_Change
12. Confidence (all the neighboring nodes) = No_Change
13. **Else**
14.    Blackhole node(s) exist
15.    Status (RREP Route) = Threat
15.    Confidence (nodes that sent RREP) = X
16.    Test threat Route (Algorithm 2)
17. **End if**

---

---

**Algorithm 2:** Test Threat Route

---

**Begin**

$k$ is the total number of successful transactions till the current data transmission instant.

**Procedure 1:**
1. $RTT_i$ (instant RTT of the route we are testing)
2. $RTT_a = \sum_{b=1}^{k} RTT_{ib}/k$
4. **If** $(RTT_i \leq RTT_a)$
5.   **Then**
6.     Examine RREP for $r$ = 1, 2, . . . , $n$
7.    **Switch** (r)
8.    **case (1)** !if RREP returns from neighboring node
9.         This is a blackhole node,
10.         Confidence (node) = −1
11.    **case (2)** !if RREP returns from not neighboring node(s)
12.         Neighboring node might be victim of a blackhole node,
13.         Confidence (nodes in the current path) = 0
14.    **Default** !$r \geq 3$
15.         Suspected node is far from the testing node, and its confidence = −1
16.         Confidence (remaining nodes in the path) = No_Change
17.     **End switch case**
18.   **Else**
19.    Suspected node is far from the testing node, and its confidence = −1
20.    Confidence (remaining nodes in the path) = No_Change,
21.    Congestion at the nodes of the current path likely exists.
22. **End If**

---

---

**Algorithm 3:** Route Selection

---

**Begin**
1. Broadcast RREQ
2. Sort received routes from AOMDV based on their minimum RTT in ascending order in an array *A*
3. While (RREP = true)
4. **If** Confidence (nodes in the RREPs) > 0
5. **Then**
6. Send the data packets through the route with the minimum RTT in array *A*
7. **Else If** Confidence (nodes in the RREPs) = 0
8. Send the data packets through the route with the minimum RTT in array *A*
9. **Else**
10. Terminate the process
11. **End if**

---

## 4. Results and Evaluation

Here, we present our experiment results for the EBR mechanism. This protocol is developed through using the AOMDV as the reactive protocol paired with our TR algorithm to detect the blackhole nodes. The performance metrics used in the simulation experiments are as follows [45–47].

### 4.1. Performance Metrics

Here, we compare the performance of the EBR protocol with AODV, AOMDV, and SUPERMAN. The performance metrics include the average energy consumption, end-to-end delay, frame delivery ratio and throughput. In addition, we simulated multiple network scenarios with and without blackhole attack. This enabled us to study the performance and effect of blackhole attacks on the other routing protocols.

End-to-end delay: This metric is defined as the average time taken for data packets transmission across the network from source to destination. This is calculated as:

$$\text{End} - \text{to} - \text{end delay} = \frac{\sum_{i=0}^{n} R\text{i} - S\text{i}}{l} \quad (1)$$

In this equation, *l* stands for the number of successfully received packets; Ri represents the current time the destination node received the *i*th packet; and Si stands for the current time the source node sent the *i*th packet.

Energy consumption: This metric is used to indicate the variations of energy consumed in the nodes.

$$\text{Energy Consumption} = \sum \text{int}_i - \text{ene}_i \quad (2)$$

In this equation, $\text{int}_i$ is the initial energy of the node *i*, and $\text{ene}_i$ is its energy at the end of the simulation.

Packet delivery ratio (PDR): Represents the percentage of the successfully received frames to the number of frames sent.

$$\text{PDR} = \frac{\text{number of packets received}}{\text{number of packets sents}} \times 100 \quad (3)$$

Throughput: This is defined as the rate of the successful data transmission through the network.

$$\text{Throughput} = \frac{\text{number of bytes Received} * 8}{\text{Simulation Time}} \times 10^6 \quad (4)$$

Routing overhead ratio: This stands for the number of routing packets needed to be sent out during the route discovery and route maintenance processes. Routing and data packets must mostly utilize the same link bandwidth. As a result, routing data is counted

as overhead. Routing overhead influences negatively the bandwidth effective use and energy consumption of the network nodes. The formula for routing overhead is as follows:

$$\text{Routing overhead } (\%) = \frac{R_p}{R_p + D_p} \times 100 \qquad (5)$$

In this equation, $R_p$ and $D_p$ stand for the number of routing packets and the number of data packets, respectively.

### 4.2. Simulation Evaluation

In Table 1, we show the assumptions of the simulation parameters. The simulation area considered is 500 m × 500 m, as we are using MANETs. Our implementation was carried out using the NS2.35. TTL has the value $n = 2$, unless otherwise stated directly.

**Table 1.** Simulation parameters.

| Parameters | Values |
| --- | --- |
| Number of Nodes | 100 |
| Mobility Speed | 10 m/s |
| Mobility | Random Way point Model |
| Propagation model | Free space propagation model |
| Area | 500 m × 500 m |
| Blackhole nodes | 10% |
| MAC Type | 802.11 |

In Figure 2a, with increasing the number of nodes, routes will have more nodes that data packets should go through. Therefore, the processing and queuing times will enlarge, particularly when routes have congested traffic. EBR outperforms other protocols, and it has the smallest time delay. In this test, the TR algorithm is not used, as there is no malicious node in the network. On the other hand, EBR selects the route with the minimum RTT. Therefore, EBR can decide the path that has the least traffic, avoiding congested routes.

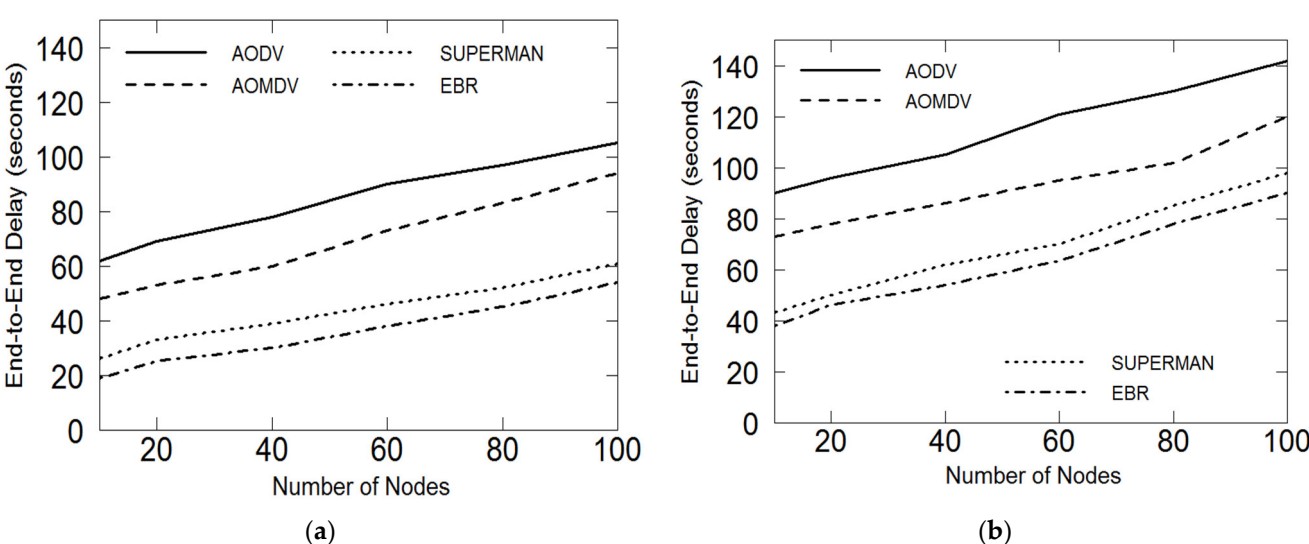

**Figure 2.** (**a**) End-to-end delay vs. number of nodes (without attack); (**b**) End-to-end delay vs. number of nodes (with attack).

In Figure 2b, the number of nodes increases, the number of blackhole nodes in the network increases, and this is because their percentage is 10%, as shown in Table 1. EBR

behaves better than other protocols when blackhole attackers exist. The AOMDV protocol returns routes in which EBR applies its TR mechanism to select those safe paths, avoiding the single and cooperative blackhole nodes. Next, EBR selects the route with the minimum RTT out of those safe paths. Therefore, EBR can determine the safe path that has the least traffic. In the case of having blackhole nodes in the path, the amount of successful data arrived at the destination ($l$ in Equation (1)) reduces due to packets drops for all tested protocols. This obviously increases the end-to-end time delay. This also happens with EBR but obviously with less harm than other protocols. EBR can detect most of the blackhole nodes, particularly neighboring and nodes close to the source node; however, further attackers would be hard to avoid.

In Figure 3, the network has 100 nodes during the whole simulation time. In Figure 3a, there are missing data packets because of the network's nature of being wireless, where data can be dropped and damaged because of the congestion, collision, etc., over time. However, as EBR selects routes based on the minimum RTT, its behavior is better than other protocols.

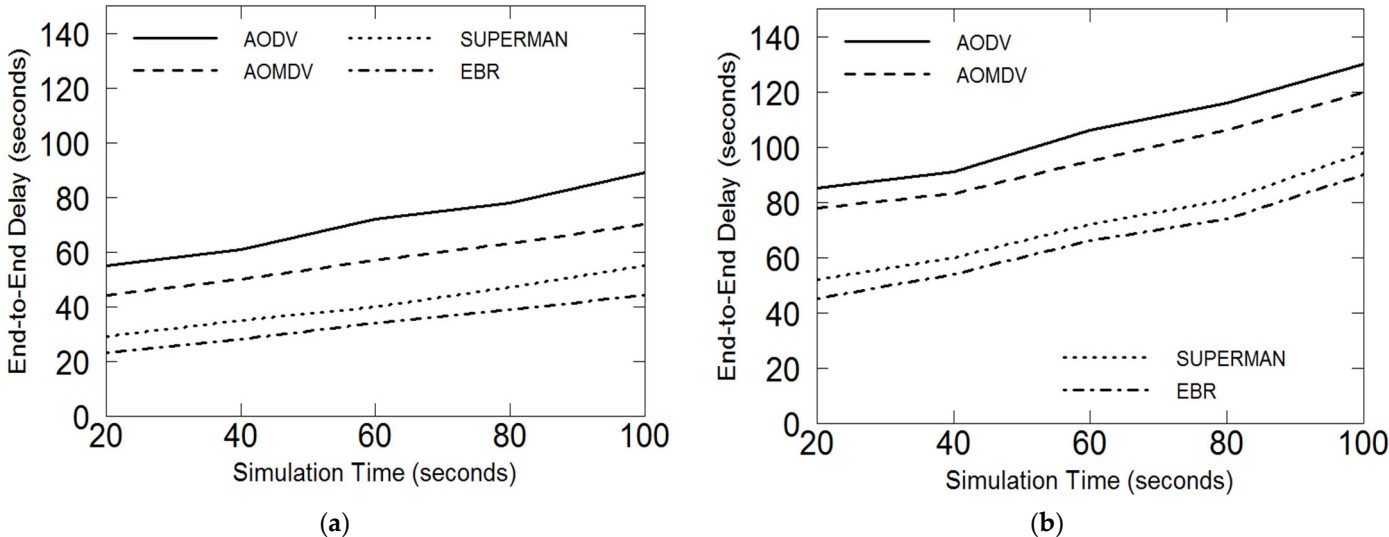

**Figure 3.** (**a**) End-to-end delay vs. simulation time (without attack); (**b**) End-to-end delay vs. simulation time (with attack).

In Figure 3b, EBR applies both its TR mechanism and minimum RTT to select the safest route with the least congestion. EBR obviously performs better than other protocols, as blackhole attackers can be detected and, accordingly, data packets follow paths avoiding suspicious nodes. On the other hand, SUPERMAN performs better than AODV and AOMDV, as it uses encryption and network layer security to prevent malicious nodes; however, it is incapable of detecting blackhole attacks as efficiently as EBR does.

SUPERMAN uses very complicated cryptographic and key-exchange mechanisms to maintain the data's safety, and this consumes a significant amount of energy, particularly in Figure 4b compared to Figure 4a. On the other hand, EBR uses less overhead in the TR mechanism to identify and resist blackhole attacks. Additionally, EBR uses the concept of minimum RTT to select the shortest route among those safe paths. This in turn would reduce the energy consumption and save the power of the nodes. This implies the smallest number of nodes and also the least congested routes.

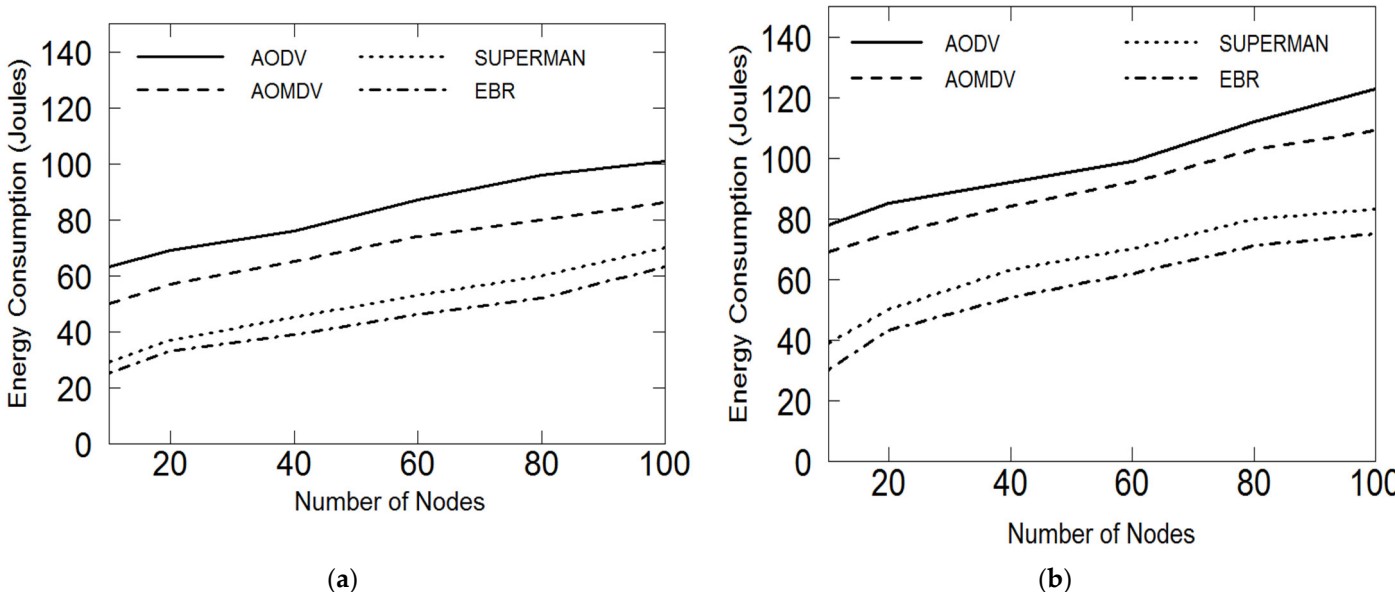

**Figure 4.** (**a**) Energy vs. number of nodes (without attack); (**b**) Energy vs. number of nodes (with attack).

In Figure 5a,b, we observe that at the range where number of nodes is greater than 80, PDR becomes almost constant and packets dropping are minimal. At this range, nodes are close to each other, and, therefore, the random loss of packets will be minimal. Consequently, the delivery ratio will be stable according to Equation (3). When the number of nodes is 100, EBR can achieve the highest PDR at 52% and 58%, with and without attack, respectively, whereas PDR for SUPERMAN is somewhat close to 40% and 50%, with and without attack, respectively. EBR uses its TR mechanism to detect the blackhole nodes and selects the legitimate path with the minimum RTT. This reduces the packets drops and maximizes the PDR sufficiently compared to other protocols.

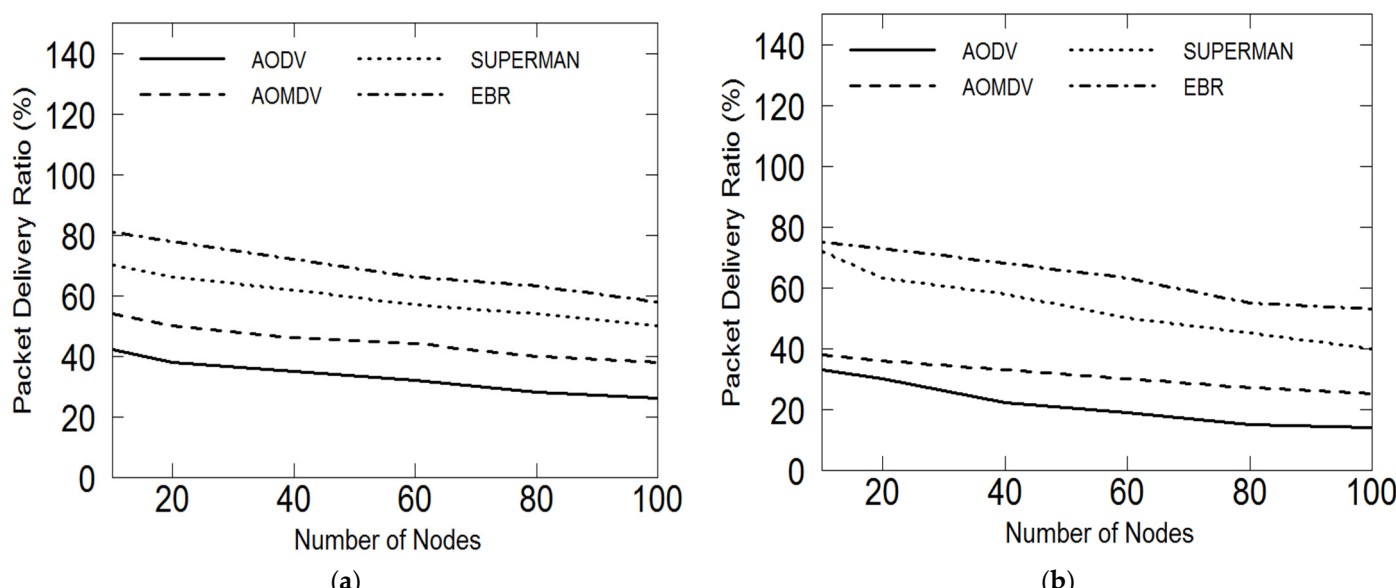

**Figure 5.** (**a**) PDR vs. number of nodes (without attack); (**b**) PDR vs. number of nodes (with attack).

In Figure 6a, EBR performs better than other protocols because the path selection is based on the minimum RTT rather than least number of nodes used by other protocols. This implies less congested routes, which in turn reduce the possibility of packets drops and, therefore, enhances the throughput. In Figure 6b, EBR gets a higher throughput gain of 7%, 60%, and 180% compared to SUPERMAN, AOMDV, and AODV, respectively. This

shows the efficiency of the EBR protocol, where it can achieve higher data delivery even with an increasing number of nodes in a MANET.

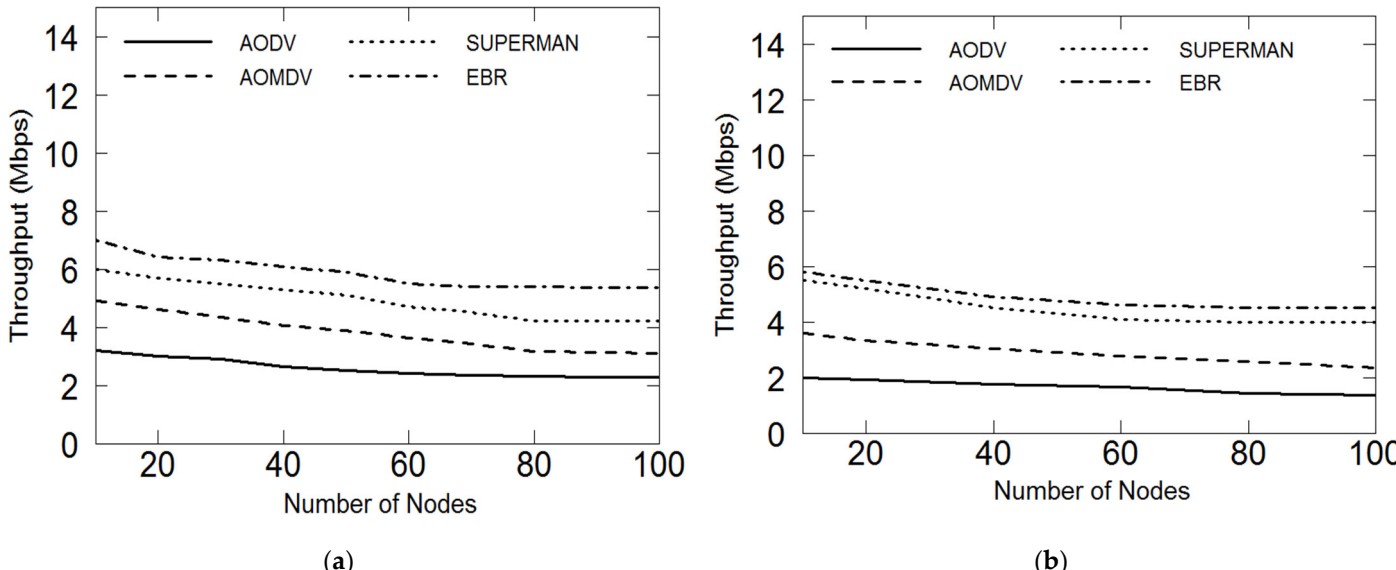

(**a**)

(**b**)

**Figure 6.** (**a**) Throughput vs. number of nodes (without attack); (**b**) Throughput vs. number of nodes (with attack).

In Figures 7 and 8, EBR behaves better than other protocols, particularly with a higher percentage of blackhole nodes. This is more obvious when EBR is compared with AODV and AOMDV protocols. Moreover, the performance gain of the EBR is about 5% compared to SUPERMAN. This shows that with increasing the number of blackhole nodes, other protocols have difficulty detecting these nodes, whereas, on the contrary, the EBR protocol uses our TR mechanism efficiently to avoid routes having blackhole nodes.

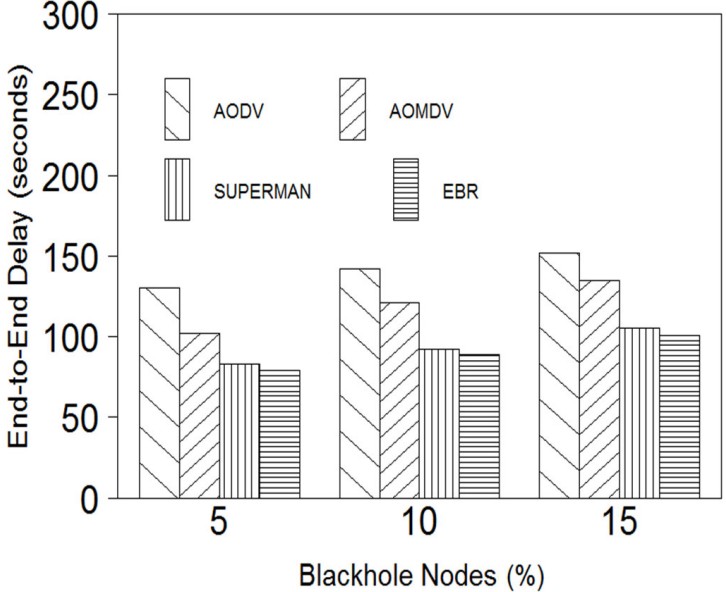

**Figure 7.** End-to-end delay vs. blackhole nodes percentage.

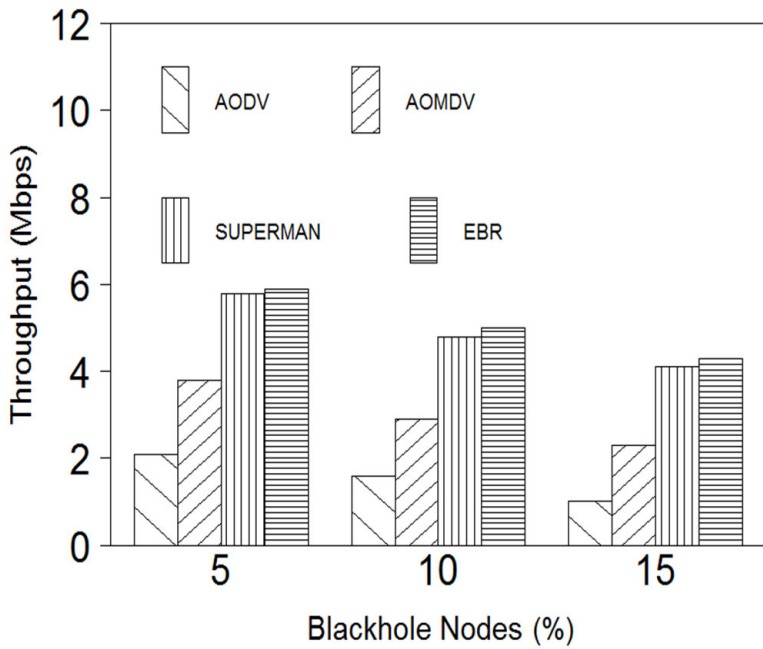

**Figure 8.** Throughput vs. blackhole nodes percentage.

In Figure 9 and when TTL increases, the source node is supposed to test more nodes in a route, and this obviously takes more time, which in turn would enlarge the end-to-end delay regardless of having blackhole nodes or not.

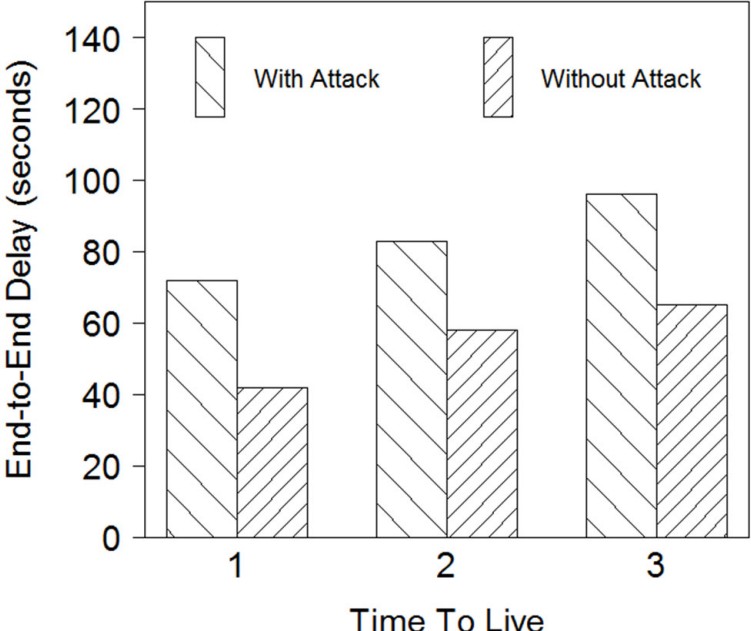

**Figure 9.** End-to-end delay vs. time to live.

In Figure 10, TTL has no impact on the PDR when there are no blackhole nodes in the network. On the contrary, when having 10% blackhole nodes in the network, and as TTL increases, the PDR improves because EBR is able to detect more blackhole nodes in the network through our TR mechanism. This is more obvious when TTL is 2 rather than being 1. PDR is indifferent when TTL = 3, as the confidence level does not change according to algorithm 2. EBR has a similar performance in Figure 11.

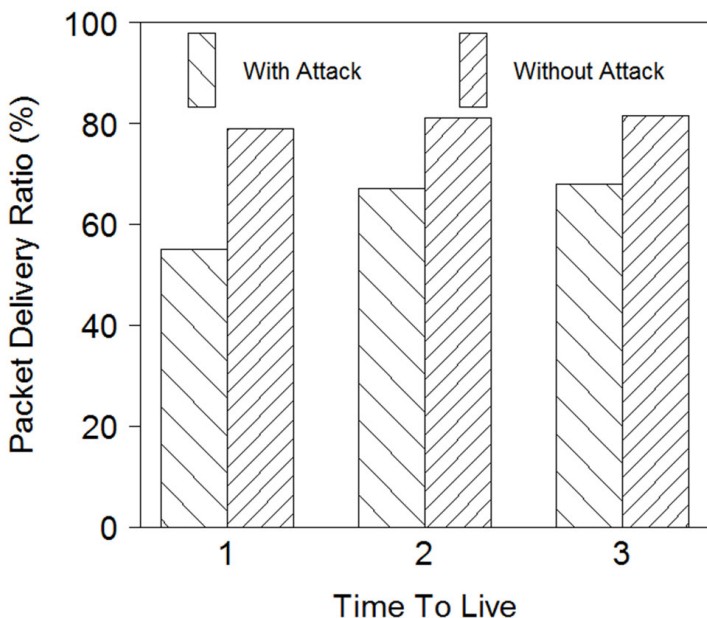

**Figure 10.** Time to live vs. PDR.

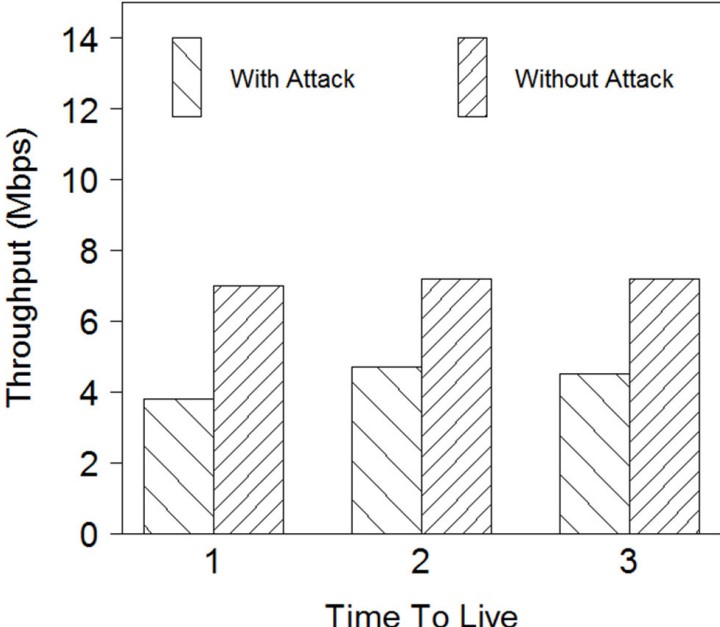

**Figure 11.** Time to live vs. throughput.

In Figure 12, the throughput increases compared to the case of having 10% blackhole nodes shown in Figure 11. At a low blackhole nodes proportion, the source node will consider the path safe because the malicious nodes don't exist close enough and, therefore, can't be detected. Accordingly, the source node considers this path as secure; however, the blackhole nodes may be located farther away. In this case, data packets will be forwarded through this unsafe path where blackhole nodes drop the packets and, therefore, the throughput is relatively low. When the proportion of these suspicious nodes increases, they likely exist close to the source node and in turn can be detected, avoiding the whole path, and, accordingly, the throughput increases, as a lower number of packets will be dropped.

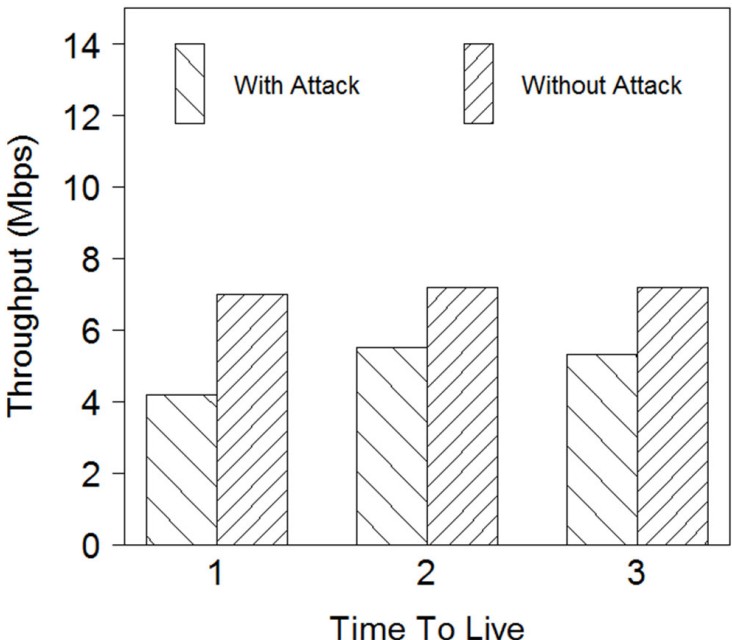

**Figure 12.** TTL vs. throughput with 30% blackhole nodes.

In Figure 13, as TTL increases, the energy consumption increases as well. EBR will heavily use the TR mechanism when TTL is large, regardless of whether there are blackhole nodes or not. This is a limitation of EBR, where small TTL is recommended to avoid such excessive energy consumption.

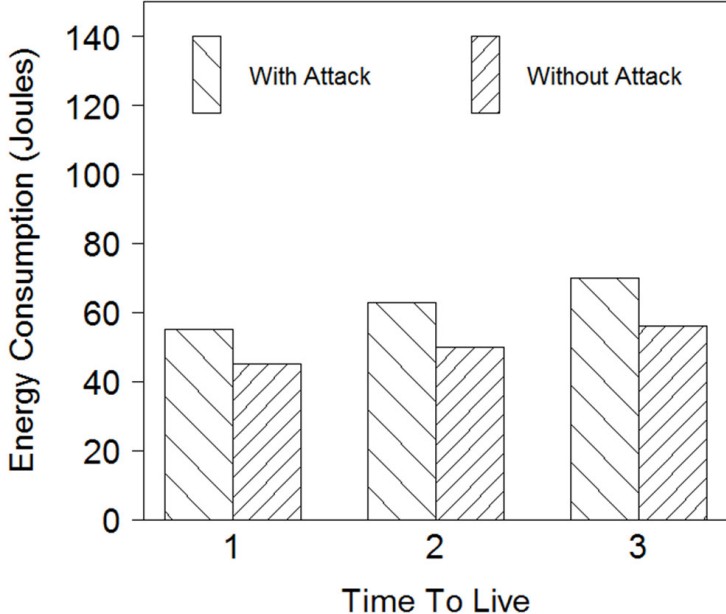

**Figure 13.** Time to live vs. energy.

One of the main advantages of the wireless networks is the ease of nodes mobility. Frequent mobility would result in data loss [48]. As in Figure 14, when nodes mobility increases, the throughput would degrade. On the other hand, EBR still outperforms other protocols with 10% blackhole nodes.

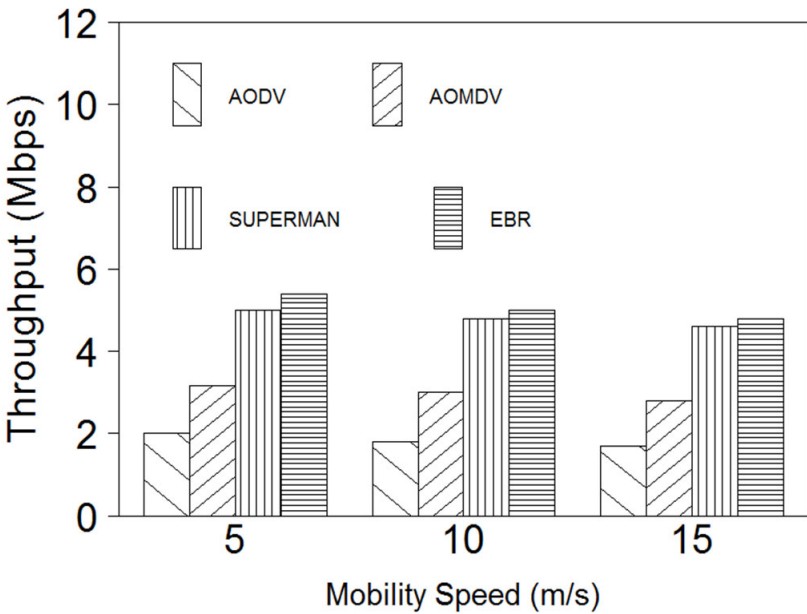

**Figure 14.** Throughput vs. mobility speed (with attack).

In Figures 15 and 16, routing overhead presented is calculated using routing packets required to broadcast requests for route discovery and maintenance. AODV and AOMDV have no mechanism to detect or resist malicious nodes; therefore, routing overhead is minimal. SUPERMAN exhibits high overhead, as it uses a complex cryptography mechanism, which in turn increases the routing packets. On the other hand, EBR uses a TR mechanism to detect and avoid blackhole nodes; however, its routing overhead is slightly higher than AOMDV. This shows the efficiency of the EBR protocol.

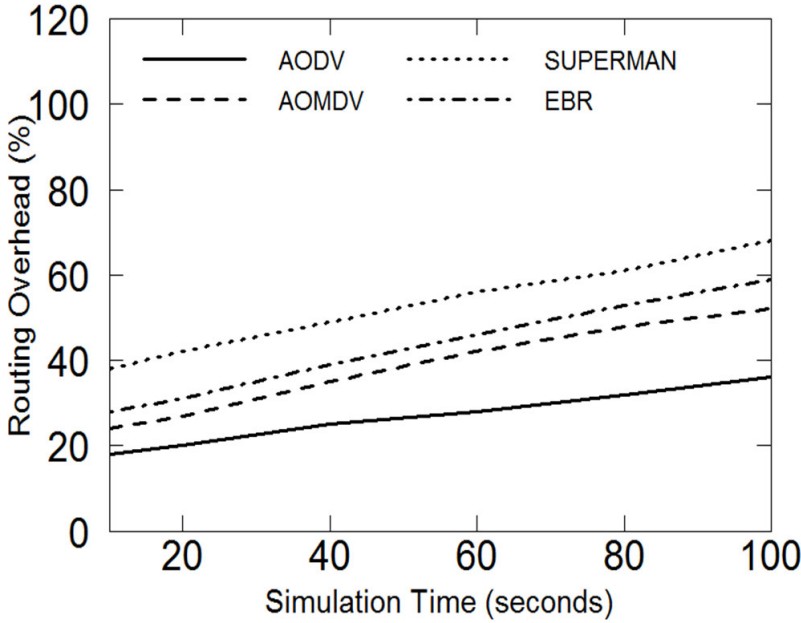

**Figure 15.** Routing overhead vs. simulation time (with attack).

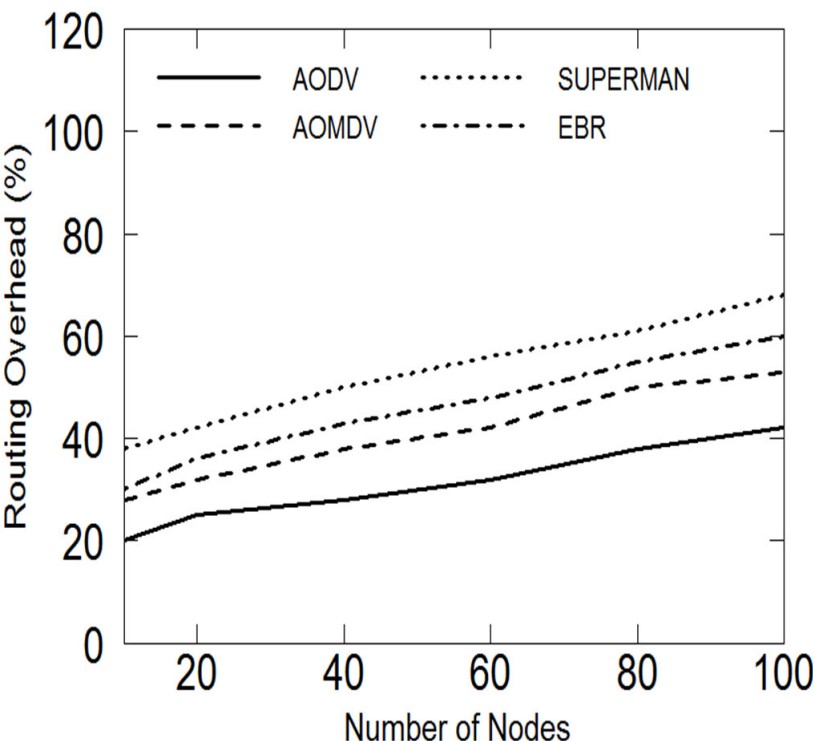

**Figure 16.** Routing overhead vs. number of nodes (with attack).

## 5. Protocol Complexity

In implementation of reactive standard routing protocols such as AODV, three methods, including Snooping, Kernel modification, and Netfilter [49], are required. Our EBR mechanism utilizes the AOMDV protocol, in which it is a multipath version of the AODV standard. Therefore, our proposed mechanism necessitates similar routing space as the case of AOMDV implementation. Other protocols are similar, such as SUPERMAN [27]. However, SUPERMAN has a lot of overhead when the number of nodes increases.

Algorithms 1 and 2 will be employed and run periodically to filter out the routes returned by AOMDV to avoid data transmission through suspicious routes (passing though blackhole nodes) or congested routes. If routes to the destination node are not available at the sender end, a route discovery process will be initiated though triggering the AOMDV protocol together with Algorithm 3, and hence multiple reliable routes will be provided. Filtering routes using EBR will not affect the processing time of the route selection using AOMDV. On the other hand, data communication through those reliable routes will save time, obviously, through reducing the data retransmissions. This was shown in the simulation results compared to other mechanisms, where network performance is enhanced.

## 6. Conclusions

The paper introduced a new concept that is able to detect and prevent malevolent intruders and also selects less congested routes. This is accomplished through a protocol that behaves normally to lure a malicious node to give a sign of its suspicious behavior. We presented an Enhanced Blackhole Resistance (EBR) protocol that can be integrated into any reactive routing mechanism in MANETs. Each node can determine the confidence level of neighboring nodes through employing a combination of time to live (TTL) and round trip time (RTT), and this in turn can determine the existence of the blackhole node(s). Out of the safe available routes, EBR would select the path with the minimum RTT, which implies the least congested routes. Our introduced protocol does not require the use of cryptographic methods, and this in turn preserves energy and computational resources. Additionally, EBR does not demand any special packets, and, therefore, the routing overhead is minimal.

Simulation results showed that the EBR protocol provides an obvious improvement of the network performance compared to AODV, AOMDV, and SUPERMAN. The proposed technique is successful in detecting blackhole nodes in a limited time, irrespective of the number of malicious nodes in a route and the time they are joining the network.

**Author Contributions:** Conceptualization, D.K. and H.E.-O.; methodology, D.K. and H.E.-O.; software, D.K.; validation, D.K.; formal analysis, D.K. and H.E.-O.; investigation, D.K. and H.E.-O.; resources, D.K.; data curation, D.K.; writing—original draft preparation, D.K.; writing—review and editing, H.E.-O.; visualization, D.K.; supervision, H.E.-O.; project administration, H.E.-O.; funding acquisition, No funding. All authors have read and agreed to the published version of the manuscript.

**Funding:** This research received no external funding.

**Conflicts of Interest:** The authors declare no conflict of interest.

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
