# Peer review of "EBR: Routing Protocol to Detect Blackhole Attacks in Mobile Ad Hoc Networks"

_electronics, doi:10.3390/electronics11213480_

Round 1

Reviewer 1 Report

Authors are suggested

1.Add Comparative results with more evaluation parameters also add future scope more clearly.

Author Response

Add Comparative results with more evaluation parameters also add future scope more clearly.

In our results, we compared our protocol with other methods using the standard parameters used in several references such as examples in references [45, 46, 47]. We have added a sentence in page 7, section 4: “The performance metrics used in the simulation experiments are as follows [45, 46, 47].”

In the introduction, we added a paragraph summarizing our contributions “In this regard, our contributions…through simulation.”

Reviewer 2 Report

  In the paper, the authors proposed an Enhanced Blackhole Resistance (EBR) algorithm based on TR mechanism for MANET  to detect and resist the blackhole attacks . Simulation results prove that EBR behaves better than other protocols. The topic is interesting and the paper is well organized. However, there are several issues need to be addressed furthermore. Please find my detailed comments as follows:

1. In the abstract, please highlight the specific metrics that the proposed methods outperform existing routing protocols. For example, in terms of……

2. In P4, please check“3.2. Problem Solution”, it is somewhat confusing, its better to change it to “proposed solution”

3. In section 3.2 , regarding “In EBR algorithm, each node can send a RREQ test message to a dummy destination node. If the receiving node(s) reply with RREP message, this will be considered as a blackhole node(s) and the confidence level will be low and therefore, the route will not be selected”,  my question is if the adversary node doesn’t reply RREP upon receiving RREQ,the proposed solution will not recognize the existence of the blackhole attacker? 

4. In section 3.3 regarding “Each node in the network periodically sends a Test_RREQ message ” , Ho do you think the resulted overhead and scalability of the solution in a large scale network? 

5. In P8, regarding algorithm1, algorithm 2, and algorithms3, it is better to make a complexity analysis and compare with existing routing protocols.   

6. In section 4.2,  the AODV is used as a benchmark for evaluation, given that SADOV is more secure, why not choose SADOV for comparison? It's better to compare the protocol with built-in security features to highlight the merit of your approach to handling security issues. 

7. Please supplement additional experiment results to show the performance of blackhole nodes detection in terms of accuracy, and time, since accurate detection is the precondition to solving the routing issue. 

8. The overall writing needs to be checked and revised carefully.

Author Response

  1. In the abstract, please highlight the specific metrics that the proposed methods outperform existing routing protocols. For example, in terms of……

Added

  1. In P4, please check“3.2. Problem Solution”, it is somewhat confusing, its better to change it to “proposed solution”

Change has been made.

  1. In section 3.2 , regarding “In EBR algorithm, each node can send a RREQ test message to a dummy destination node. If the receiving node(s) reply with RREP message, this will be considered as a blackhole node(s) and the confidence level will be low and therefore, the route will not be selected”,  my question is if the adversary node doesn’t reply RREP upon receiving RREQ,the proposed solution will not recognize the existence of the blackhole attacker? 

This is right, so if no RREP is received, this confirms that neighboring nodes of the adversary node are not blackhole attackers.

  1. In section 3.3 regarding “Each node in the network periodically sends a Test_RREQ message ” , Ho do you think the resulted overhead and scalability of the solution in a large scale network?

Figure 18 shows the effect of network scalability on the overhead. It shows that with a larger network, the overhead increases but in a less way than other protocols.

  1. In P8, regarding algorithm1, algorithm 2, and algorithms3, it is better to make a complexity analysis and compare with existing routing protocols.

Section 5 is added.

  1. In section 4.2,  the AODV is used as a benchmark for evaluation, given that SADOV is more secure, why not choose SADOV for comparison? It's better to compare the protocol with built-in security features to highlight the merit of your approach to handling security issues. 

In reference [27]. it was proved that SAODV requires more complex routing behaviour than AODV and SUPERMAN protected AODV (SUPERAODV). SAODV and SUPERAODV generate a security overhead of 66.6 and 36.9 percent more bytes, when compared to AODV in networks of 100 nodes, respectively. Therefore, we use AODV and SUPERMAN as benchmark for comparisons with EBR.

  1. Please supplement additional experiment results to show the performance of blackhole nodes detection in terms of accuracy, and time, since accurate detection is the precondition to solving the routing issue. 

In our results, we compared our protocol with other methods using the standard parameters in several references such as examples in references [45, 46, 47]. We have added a sentence in page 7, section 4: “The performance metrics used in the simulation experiments are as follows [45, 46, 47].”

  1. The overall writing needs to be checked and revised carefully.

Writing is checked.

Reviewer 3 Report

The article's technical soundness is excellent and timely. However, the authors should compare the performance metrics, such as Average energy consumption, End-to-End latency, Frame Delivery Ratio, and Throughput, with the work presented by the other authors mentioned in the literature review.

Second, the authors should revise the work's contributions and motivations.

Author Response

  1. The article's technical soundness is excellent and timely. However, the authors should compare the performance metrics, such as Average energy consumption, End-to-End latency, Frame Delivery Ratio, and Throughput, with the work presented by the other authors mentioned in the literature review.

Our work is compared with AODV [16], AOMDV [3] and SUPERMAN [27]. SUPERMAN is a recent secured protocol.

  1. Second, the authors should revise the work's contributions and motivations.

In the introduction, we added/amended paragraphs to revise the motivations and contributions.

Round 2

Reviewer 3 Report

The authors have responded to some of my comments. I believe the authors need to slightly strengthen the conclusion, which is currently inadequate. In addition, the formatting and structure of the article require improvement. Since the authors have covered the majority of the relevant literature, just a few additional publications must be considered such as (https://www.mdpi.com/2504-446X/2/3/27) for literature review and https://ieeexplore.ieee.org/abstract/document/9915455, to make the introduction section better.

Author Response

The conclusion has been amended.

Although those articles that the reviewer has advised to be added are of a high quality papers, however, they are irrelevant to our manuscript.